# Design of an Electromagnetic Linear Drive with Permanent Magnetic Weight Compensation

Bela Schulte Westhoff and Jürgen Maas *

Mechatronics Systems Laboratory, Technische Universität Berlin, 10785 Berlin, Germany;
schultewesthoff@tu-berlin.de
* Correspondence: juergen.maas@tu-berlin.de

**Abstract:** When using electric linear drives for vertical positioning of workloads, a constant force both during movement and at standstill must be supplied to compensate gravity. Compensating stationary forces by the use of passive components reduces the power consumption of the employed actuator and permits smaller dimensioning. In this article, we present a novel integrative actuator design which combines the inherent advantages of a permanent magnetic weight compensation with a two-phase linear direct drive. We illustrate how to design permanent magnetic force compensation to realize a constant compensating force over a desired actuator stroke. Analytical solutions for both the design of the direct drive and the design of the permanent magnetic weight compensation are derived and validated by simulation and experiment. The innovative actuator design is compared to a conventional, non-compensated drive, and we aim to provide the reader with insights into specific applications where the use of the weight-compensated actuator proves particularly effective.

**Keywords:** magnetic weight compensation; miniaturized linear drive; two-phase direct drive





## 1. Introduction

To compensate the gravitational forces of the workload when using electric linear direct drives for single-axis vertical positioning, a constant force both during movement and at standstill must be supplied. Unlike active force control technology, such as magnetic bearings which necessitate additional sensors, demanding control algorithms, and sophisticated hardware configurations [1], the integration of passive components reduces the power consumption of the electric actuator and permits smaller dimensioning without significantly increasing system complexity and cost. Various passive gravity compensation mechanisms exist, with commonplace solutions including counterweights or mechanical frameworks employing diverse combinations of springs, sliders, and beam structures [2–7]. Despite the wide variety of options, these structures frequently encounter limitations such as challenges related to compactness within constrained design spaces, restricted constant-force stroke ranges, pronounced output errors, or demanding manufacturing requirements [6].

Beyond the conventional mechanical approaches, permanent magnetic springs constitute another alternative for weight compensation. Magnetic springs are widely employed in various applications including vibration compensation springs [8–10], torsion magnetic springs [11,12] and linear magnetic springs [13,14]. Through careful design considerations of both soft magnetic and permanent magnetic elements, magnetic spring systems can achieve a consistent force output across arbitrary process strokes, regardless of the position of the actuator's slider [13,14]. Next to the large constant-force stroke, the inherent advantage of magnetic weight compensation lies in its suitability for deployment in constrained design spaces. This feature positions magnetic weight compensation as a viable alternative to traditional mechanical frameworks for weight compensation applications.

In this paper, we propose an innovative, integrative design of a two-phase linear direct drive incorporating permanent magnetic weight compensation. The schematic design

concept of the novel actuator is illustrated in Figure 1. The two-phase linear drive forms the active part, generating dynamic force $F_{dy}$, while the weight compensation constitutes the passive part, compensating constant gravity force $F_{wc}$. Neglecting frictional forces and other dissipative disturbances, the differential equation of the actuator is given by $x_s$ denotes the position of the slider, $m$ is the load mass and $g$ represents gravity. The weight compensation is designed to compensate static force $F_{st}$, while the linear drive is intended to generate dynamic force $F_{dy}$ to accelerate the workload:

$$m\ddot{x}_s + mg = F_{wc} + F_{dy}. \tag{1}$$

$$\begin{aligned} F_{wc} &\stackrel{!}{=} F_{st} = mg, \quad \text{and} \\ F_{ld} &\stackrel{!}{=} F_{dy} = m\ddot{x}_s. \end{aligned} \tag{2}$$

To provide a measure of effectiveness of the weight compensation, we define the ratio of the static and dynamic force by introducing compensation factor $V$:

$$V = \frac{|F_{st}|}{\left|F_{dy}\right|}. \tag{3}$$

For a harmonic excitation of the workload with frequency $\omega$ and amplitude $\hat{x}_a$, compensation factor $V$ is calculated as

$$V = \frac{g}{\omega^2 \hat{x}_a} \tag{4}$$

The larger the compensation factor $V$, the more the passive compensation offloads the active actuator part and the greater the benefit of an integrated magnetic weight compensation. Given a factor of $V = 1$, for example, the needed force for the weight compensation equals that for dynamic acceleration.

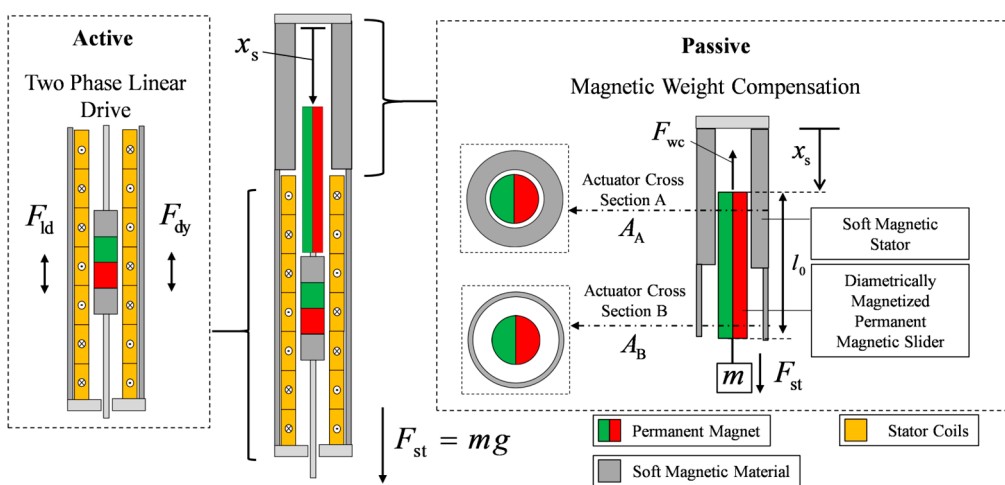

**Figure 1.** Schematic structure of the integrative actuator design of a two-phase linear direct drive with permanent magnetic weight compensation.

In this article, we present the analytical, numerical, and experimental investigation of the presented actuator. The paper is divided into two main chapters. In the first, the focus is placed on the analytical design of permanent magnetic weight compensation. A general analytical solution for the resultant force dependent on specified design parameters is derived and validated numerically. Based on this, the integration of the weight compensation into a two-phase linear direct drive is presented. In the second part, a weight-compensated actuator is designed, realized, and experimentally characterized for a selected use case. In the concluding discussion, an evaluation is given of the applications for which the integration of weight compensation is particularly effective.

## 2. Analytic Modeling of a Weight-Compensated Linear Direct Drive

*2.1. Analytic Modeling of Magnetic Weight Compensation*

The objective of weight compensation is to generate constant force $F_{wc}$ to equalize static load $F_{st}$, independent of slider position $x_s$. The general working principle of a permanent magnetic weight compensation is illustrated in Figure 1. The diametrically magnetized permanent magnet (slider) seeks the energy minimum and strives into the soft magnetic stator, thereby generating compensating force $F_{wc}$. To analytically derive the weight compensating force in dependence of the design parameters, the total magnetic energy of magnetized matter $U_m$ in the weight compensating system in dependence on slider position $x_s$ is required. The reluctance force can then be calculated using the variation of energy:

$$F_{wc} = -\frac{dU_m(x_s)}{dx_s}. \tag{5}$$

To calculate $U_m$ of the regarded system illustrated, the magnetic field distribution needs to be determined. To find an approximate analytic solution, we impose the following restrictive assumptions on the system:

- neglect axial magnetic flux,
- assume constant diametrical magnetization of the permanent magnet,
- regard the soft magnetic material as an ideal magnetic conductor,
- neglect saturation effects.

The total energy of weight compensation $U_{wc}$ is derived in Appendix A. The energy is dependent on the magnetization of permanent magnet $M_{pm}$, the cross-sections of permanent magnet $A_{pm}$ and of soft magnetic stators $A_{sm, A}$ and $A_{sm, B}$ (Figure A1b) as well as on axial position $x_s$ and length $l_0$ of the permanent magnet,

$$U_{wc} = \mu_0 \frac{M_{pm}^2}{4} A_{pm} \left( \left( 1 - \frac{A_{pm}}{A_{sm, A}} \right) x_s - \left( 1 - \frac{A_{pm}}{A_{sm, B}} \right) (l_0 - x_s) \right). \tag{6}$$

The compensating force follows to

$$F_{wc} = -\frac{dU_{wc}}{dx_s} = \mu_0 \frac{M_{pm}^2}{4} A_{pm} \left( \frac{A_{pm}}{A_{sm, A}} - \frac{A_{pm}}{A_{sm, B}} \right). \tag{7}$$

Derived force $F_{wc}$ of the magnetic weight compensation is independent of the slider position and depends solely on the magnetization and the geometric cross-section parameters of the stator and the slider. For the sample parameters,

$$\begin{aligned} M_{pm} &= \frac{B_{Rem}}{\mu_0} = \frac{1.28T}{4\pi 10^{-7}} \approx 1.02 \times 10^6 \frac{A}{m}, \\ R_{pm} &= 1 \text{ mm} \end{aligned} \tag{8}$$

and in dependence of air gaps $d_A$ and $d_B$ between the permanent magnet and the soft magnetic stator in the two cross-sections,

$$d_A = R_{sm,A} - R_{pm} \text{ and } d_B = R_{sm,B} - R_{pm}, \tag{9}$$

the weight compensating forces are plotted in Figure 2. The plot displays the strong dependence of force on the air gap widths. To obtain maximum force density, the air gap in cross-section A should be minimized and maximized in cross-section B.

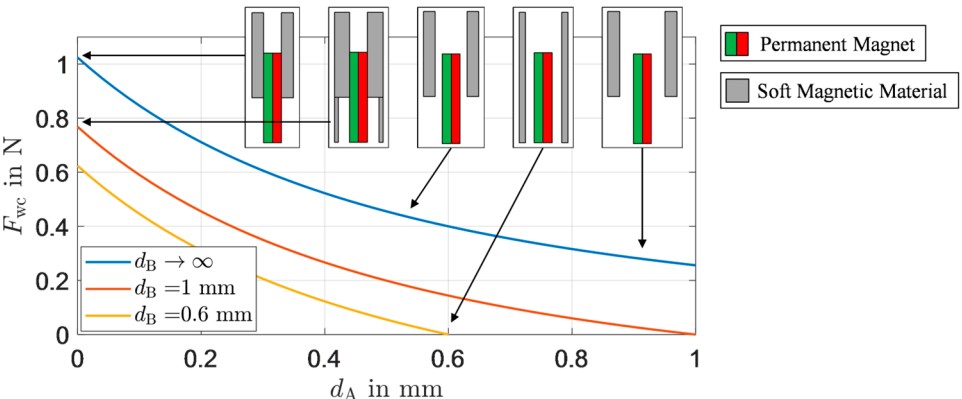

**Figure 2.** Weight-compensating force in dependence of air gap widths.

In order to derive an analytical formula for the compensating force, we make some assumptions to simplify the regarded system. By neglecting axial magnetic flux, the derived force is completely independent of the slider position and the axial length of the weight compensation. In the following, we validate the analytical solution with a numerical FEM simulation for an example set of parameters. Next to (8), we specify

$$r_{\text{sm, A}} = 1.1 \text{ mm}, \ l_{\text{pm}} = 30 \text{ mm}. \tag{10}$$

The analytic and numeric solutions of the compensating force and magnetic energy are illustrated in Figure 3. The FEM simulation reveals that the compensating force is not independent of the slider position. Especially in the range in which the permanent magnet enters and exits the soft magnetic stator, axial flux linkage has a significant influence on the compensating force. Consequently, when designing permanent magnetic weight compensation, axial range should be considered, in which the analytical force is not matched. For most of the actuator stroke, the analytical solution offers a very accurate approximation of the compensating force.

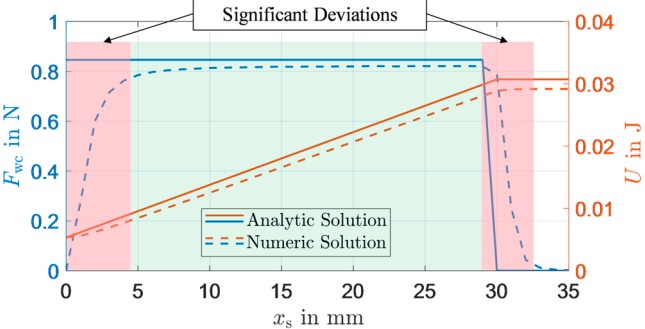

**Figure 3.** Stored magnetic energy and compensating force in dependence of the slider position.

### 2.2. Analytic Modeling and Integration of the Two-Phase Linear Drive

To generate dynamic forces, passive weight compensation needs to integrated into an active linear drive. In this work, we combine weight compensation with a two-phase linear direct drive, which is illustrated in Figure 4. The two-phase linear direct drive generates high force density, which renders it particularly suitable for actuators characterized by stringent spatial constraints, as demonstrated in reference [15]. The axially magnetized permanent magnets in the slider generate a magnetic field which permeates the stator coils in radial direction. An axial Lorentz force is thus generated by energizing stator coils $\alpha$ and $\beta$ in dependence of slider position $x_{\text{s}}$. A detailed derivation of the analytical modeling and the electric control of the two-phase linear drive is given in [15]. The actuator force

amplitude, $F_\mathrm{L}$, is proportional to the current amplitude of the excitation, $\hat{i}$, and the force constant, $k_\mathrm{m}$:

$$F_\mathrm{L} = k_\mathrm{m} \hat{i} \text{ with } k_\mathrm{m} = \frac{p}{\lambda} B_\mathrm{rem} A_\mathrm{pm,a} 2\pi N . \tag{11}$$

Analytically derived force constant $k_\mathrm{m}$ increases linearly to the number of axial permanent magnets $p$ in the slider (four in Figure 4). $B_\mathrm{rem}$ is the remanent flux density of the permanent magnets, $\lambda$ is the wavelength of the radial magnetic flux density distribution, $A_\mathrm{pm,a}$ is the cross-section area of the axial magnetized magnet and $N$ offers the number of windings of each coil package.

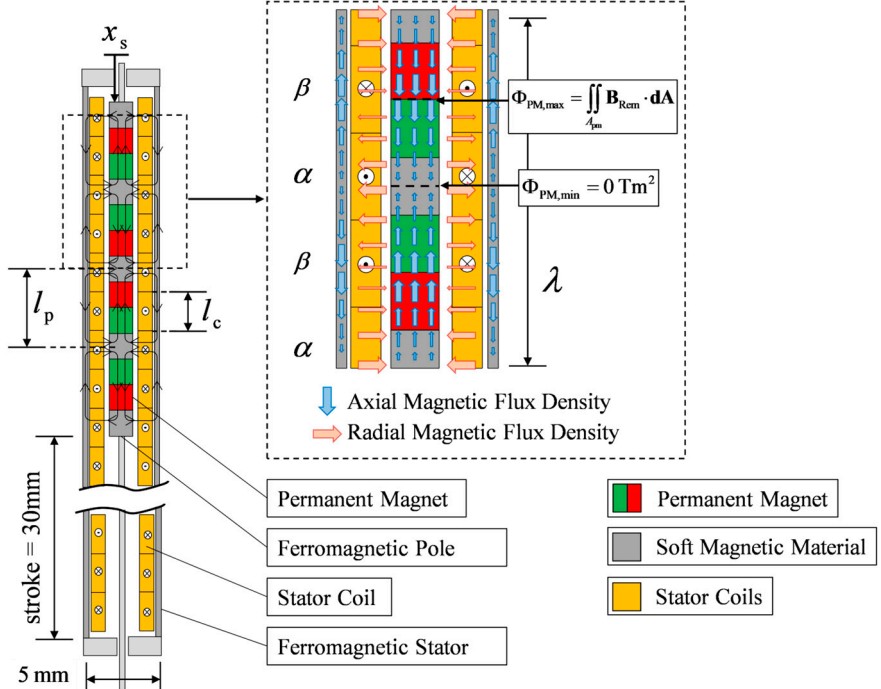

**Figure 4.** Linear direct drive equipped with a two-phase coil winding and axially magnetized magnets inside the slider.

Generally, the passive magnetic weight compensation and active linear drives as presented can be conceived and developed as separate entities. However, when integrating these components, it is imperative to assess the influence exerted by the diametrically magnetized permanent magnet inherent to the weight compensation slider upon its engagement with the active linear drive. The integration of the permanent magnetic weight compensation into the linear drive is visualized in Figure 5. The diametrically magnetized, weight compensating permanent magnet enters the current-carrying stator coils of the linear actuator without generating an interfering axial Lorentz force (middle part in Figure 5). Due to the divergence-free nature of magnetic flux density, it permeates the stator coils in equal parts in positive and negative radial directions. The resulting Lorentz forces cancel and thus do not generate any disturbing force on the slider.

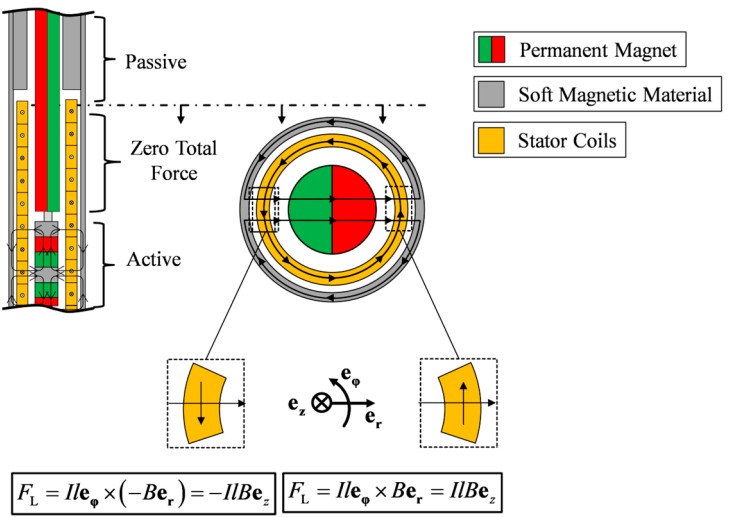

**Figure 5.** Integration of the permanent magnetic weight compensation into the linear drive [14].

## 3. Design and Experimental Validation

In this chapter, a linear actuator with integrated weight compensation is realized following the design concept developed in Section 2.

### 3.1. Application for a Weight-Compensated Linear Drive—Automated Electrical Discharge Machining

The presented actuator concept is particularly effective when the introduced compensation factor $V$ is sufficiently large. In this chapter, we design a weight-compensated actuator for a suitably selected problem with comparatively large compensation factor. The objective of the developed linear direct drive is the vertical positioning of a tool electrode in the die-sinking electrical discharge machining (EDM) process which enables precise machining of hard materials with complex shapes and structure [16]. According to the current state of the art, a countersunk head of the workpiece is manufactured at high cost using traditional mechanical processing methods [16,17]. To dynamically adapt the geometry of the countersink electrode to the complex geometries of the workpiece, a discrete tool electrode assembly must be developed [18]. As illustrated in Figure 6, this assembly demands the vertical positioning of multiple 20 g electrode rods to automatically create a discrete negative geometry of a desired workpiece. Since the width of the square electrode segments is $w_{\mathrm{E}} = 5$ mm, the miniaturized linear drive must dispose an outer diameter of $d_o \leq 5$ mm. Furthermore, a static process stroke of $s \geq 25$ mm is required to dynamically generate complex depth profiles for highly adaptable tool geometries. The major challenge in designing the actuator is to ensure sufficient force supply for robust control dynamics despite the small installation space. To reduce the load on the actuator and thereby permit smaller dimensioning, the permanent magnetic weight compensation is integrated to passively counterbalance the electrode mass. Application-related, the actuator is demanded to generate a harmonic oscillation of the electrode, for which the actuator is required to operate at a frequency of $\omega \leq 5$ Hz with an amplitude of $\hat{x}_a = 1000$ µm [18]. Accordingly, the static-to-dynamic force amplitude ratio $V$ for the presented application results in

$$V = \frac{F_{\mathrm{static}}}{F_{\mathrm{dynamic,amp}}} = \frac{mg}{m\omega^2 \hat{x}_a} \approx \frac{0.2 \text{ N}}{0.02 \text{ N}} \approx 10. \tag{12}$$

The gravitational force of the 20 g electrode exceeds the dynamic force amplitude for acceleration by a factor of 10. This makes the weight-compensated actuator a perfect solution for the presented application. Under the negligence of friction forces, the dynamic part of the actuator can be dimensioned 10 times smaller due to the integrated passive weight compensation.

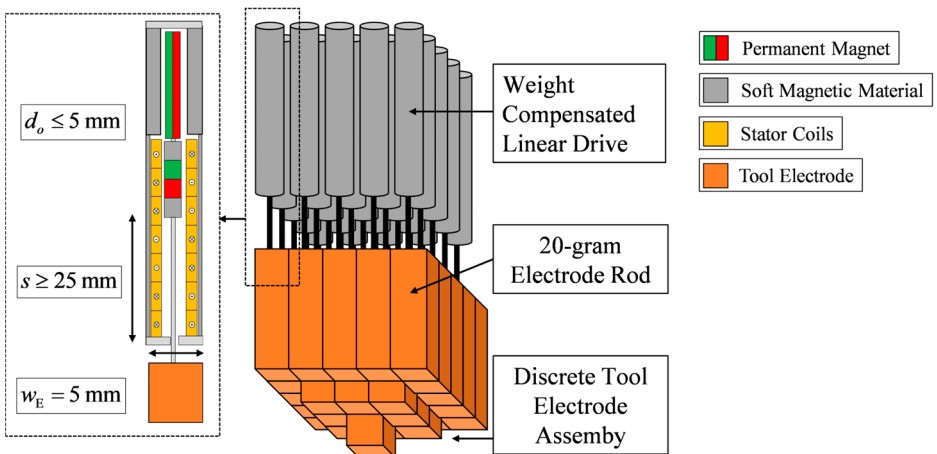

**Figure 6.** Discrete tool electrode assembly for automated tool electrode geometry generation in electrical discharge machining.

### 3.2. Design of the Weight-Compensated Linear Drive for the EDM Process

The designed and realized weight-compensated linear drive is illustrated in Figure 7. The soft magnetic stator within the actuator serves as a magnetic isolator, effectively preventing magnetic flux leakage from escaping the actuator. Radius $R_{\mathrm{sm,\,B}}$ of cross-section area $A_{\mathrm{sm,\,B}}$ of the weight compensation is predefined by the design of the two-phase linear drive, which is predefined by the limited outer diameter of the actuator, as illustrated in Figure 7:

$$A_{\mathrm{sm,\,B}} = R_{\mathrm{sm,\,B}}{}^2 \pi = (2.4\ \mathrm{mm})^2 \pi. \tag{13}$$

Parameters $R_{\mathrm{sm,\,A}}$ and $R_{\mathrm{pm}}$ of magnetic weight compensation are adaptable design parameters to obtain the desired compensation force of 0.2 N for the 20 g electrode. Following (7), we obtain

$$F_{\mathrm{wc}} = -\frac{dU}{dx} = \mu_0 \frac{M_{\mathrm{pm}}{}^2}{4} A_{\mathrm{pm}} \left( \frac{A_{\mathrm{pm}}}{A_{\mathrm{sm,\,A}}} - \frac{A_{\mathrm{pm}}}{A_{\mathrm{sm,\,B}}} \right) \overset{!}{=} 0.2\ \mathrm{N}. \tag{14}$$

Since the permanent magnet is considerably more expensive than soft magnetic material, it is reasonable to dimension the permanent magnet preferably small. We choose a permanent magnet with the following parameters:

$$R_{\mathrm{pm}} = 1\ \mathrm{mm},\ B_{\mathrm{Re}m} = 1.2\ \mathrm{T},\ M_{\mathrm{pm}} = \frac{B_{\mathrm{Re}m}}{\mu_0}. \tag{15}$$

Solving (14) for the inner radius, $R_{\mathrm{sm,\,A}}$, results in

$$R_{\mathrm{sm,\,A}} = 1.2\ \mathrm{mm}. \tag{16}$$

The parameters of the two-phase linear drive are chosen to be

$$\begin{aligned} p = 8, \quad \lambda = 8\ \mathrm{mm}, \quad A_{\mathrm{pm,a}} = 4.15\ \mathrm{mm}^2, \\ B_{\mathrm{rem}} = 1.1\ \mathrm{T}\ \ \mathrm{and}\ \ N = 8, \end{aligned} \tag{17}$$

which results in a force constant of

$$k_m = \frac{p}{\lambda} B_{\mathrm{Rem}} A_{\mathrm{pm,}l} 2\pi N = 0.208\ \frac{\mathrm{N}}{\mathrm{A}}. \tag{18}$$

The saturation current amplitude is chosen to be

$$\hat{i}_{\mathrm{sat}} = 2.5\ \mathrm{A}, \tag{19}$$

which corresponds to a current density of

$$j \approx 7 \, \frac{A}{\text{mm}^2}.$$

(20)

Considering the required dynamic force, it is evident that the dynamic part of the actuator is oversized. This is due to the fact that the dynamic part must compensate for any friction and other disturbing forces next to the acceleration force, which is investigated in the next section. To control the actuator and to locate the slider position, a Hall sensor is implemented into the actuator (more details on the measurement process are given in [19]).

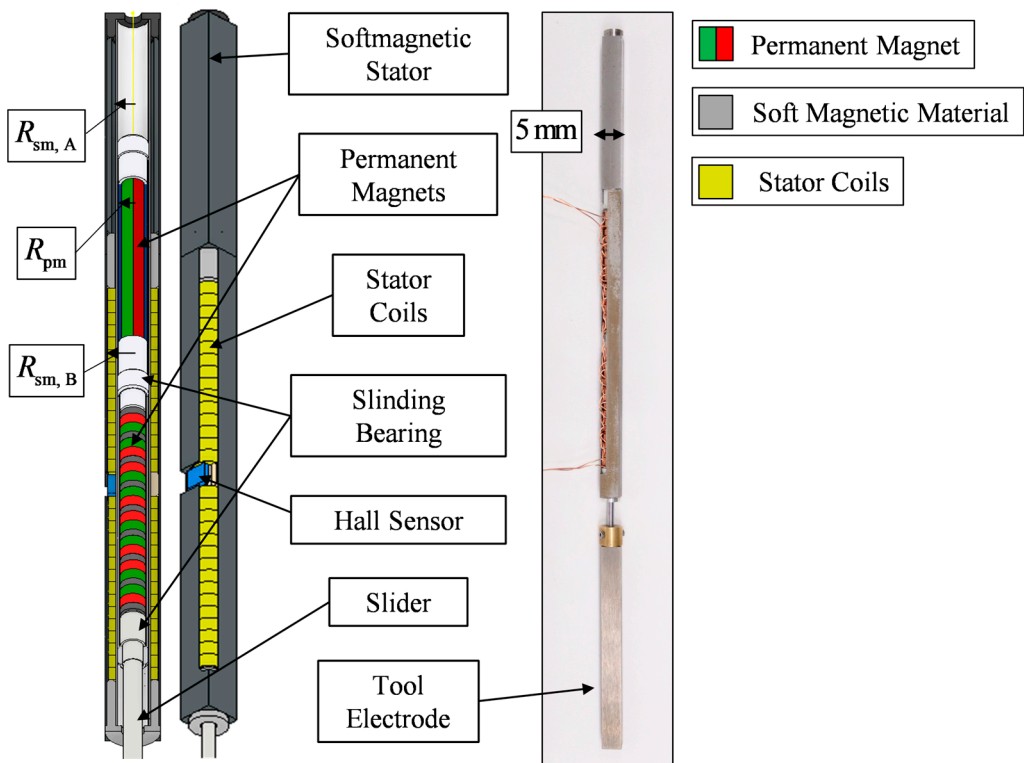

**Figure 7.** Design and realization of the weight-compensated linear drive.

*3.3. Experimental Characterization of Weight-Compensated Linear Drive*

The experimental setup for the characterization of the actuator is illustrated in Figure 8. We use an external linear drive to position the actuator slider while measuring the weight compensating force as well as the friction on the slider. Since the maximum actuator velocity at the desired amplitude and frequency is sufficiently small, we neglect the effect of velocity on friction in this consideration. The friction effects on the slider are approximated by a Coulomb friction model:

$$F_{\text{R}}(x_{\text{s}}) = \mu_{\text{r}} F_{\text{N}}(x_{\text{s}}).$$

(21)

$\mu_r$ is the friction coefficient and $F_{\text{N}}(x_{\text{s}})$ is the normal force on the slider. The normal force mainly arises due to the reluctance force between the soft magnetic stator and the permanent magnets in the slider of the actuator. Under ideal symmetry, the slider is in an unstable equilibrium in the middle of the stator. Manufacturing inaccuracies of the stator and slider violate this ideal symmetry and cause the friction-generating normal force on the slider. This effect is enhanced by inhomogenities of the soft and hard magnetic materials and production-related asymmetries of the soft magnetic stator. In order to mitigate friction, polytetrafluoroethylene (PTFE) is employed as the material for the sliding bearing.

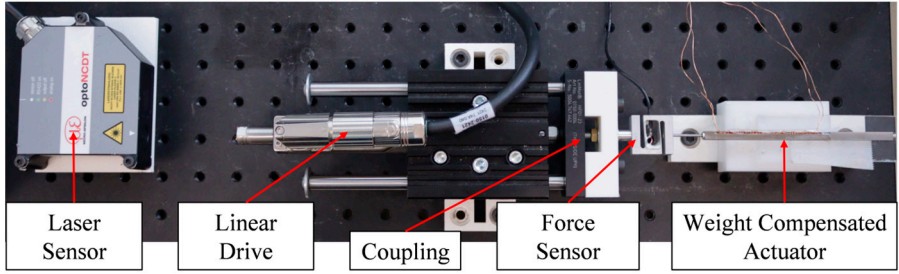

**Figure 8.** Experimental setup for the characterization of the weight-compensated actuator.

The measured weight compensating force of the actuator is shown in Figure 9a. As illustrated, the analytically designed weight compensation builds a very accurate approximation of the measured force profile. The mean of the measured compensating force deviates from the analytical solution by <5%.

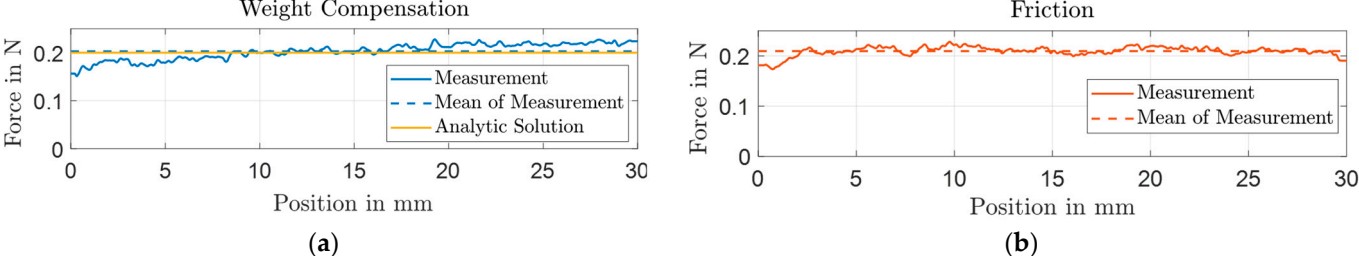

(**a**)           (**b**)

**Figure 9.** (**a**) Position-dependent force of weight compensation; (**b**) Position-dependent friction of the actuator.

For the designed actuator, friction has a very large influence on the system behavior. The position-dependent friction of the slider is determined by measuring the force–position hysteresis when sweeping the actuator stroke at constant speed. The friction is illustrated in Figure 9b and is approximated by the measured mean:

$$F_{\mathrm{R}} = 0.21 \, \mathrm{sgn}(\dot{x}_{\mathrm{s}}) \, \mathrm{N}. \tag{22}$$

Accordingly, the frictional force exceeds the required dynamic force amplitude for mere acceleration of the electrode by a factor of 10. The adapted differential equation of the actuator considering the examined friction results in

$$m_l \ddot{x}_{\mathrm{s}} - m_l g = F_{\mathrm{wc}} + k_{\mathrm{m}} i + F_{\mathrm{R}}. \tag{23}$$

All emerging dissipating forces must be compensated by the dynamic actuator part. Taking friction into account, the adapted compensating factor $\widetilde{V}$ shifts significantly:

$$\widetilde{V} = \frac{|F_{\mathrm{static}}|}{|F_{\mathrm{dynamic}}|} = \frac{mg}{m\omega^2 \hat{x}_a + F_R}$$
$$= \frac{0.2 \, \mathrm{N}}{0.02 \, \mathrm{N} + 0.21 \, \mathrm{N}} \approx 0.9 \, . \tag{24}$$

The friction force reduces the effectiveness of the weight compensation to a significant extent. The magnitude of friction in the designed actuator depends on various parameters which make it difficult to generalize the impact of friction on the compensation factor before manufacturing a desired actuator. First, the ratio of actuator stroke to actuator diameter of the linear drive needs to be considered. The friction force increases proportionally to the length of the weight compensating permanent magnet, which linearly depends on the actuator stroke. For the regarded application, the ratio is exceedingly large, leading to high friction and thus a strong decrease from $V$ to $\widetilde{V}$. Moreover, the considered actuator is miniaturized ($d_{\mathrm{a}} \leq 5$ mm), which causes absolute manufacturing inaccuracies to have a greater influence on asymmetries in relation to the size of the actuator as for larger

actuators. In general, friction depends on manufacturing inaccuracies as well as on material inhomogeneity, which makes it difficult to estimate its impact beforehand. However, an estimate on the frictional forces is required to be considered when designing an actuator. For the application under consideration, the adapted compensation factor $\widetilde{V}$ is approximately one, allowing for the active actuator part to be dimensioned half the size of that without the implemented weight compensation. This is still a significant reduction in dimension which justifies the manufacturing expense of weight compensation.

To validate the force coefficient $k_m$ of the active actuator part, we consider the dynamic force for different current amplitudes $\hat{i}$, as illustrated in Figure 10. In the linear range, force factor $k_m$ is approximated by

$$k_m \approx 0.16 \, \frac{\text{N}}{\text{A}}. \tag{25}$$

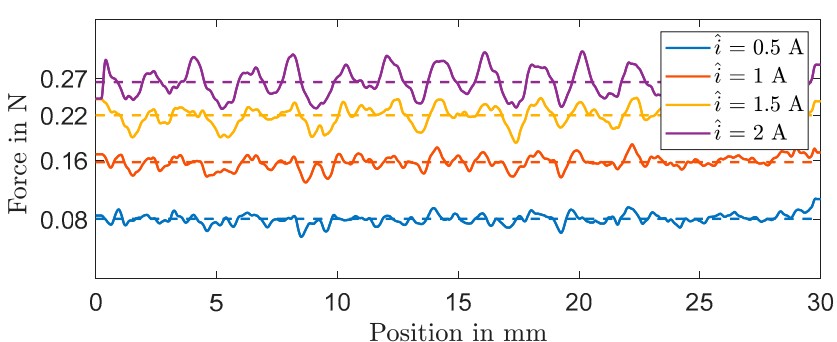

**Figure 10.** Force of the two-phase linear drive in dependence of position and current amplitude.

At current amplitudes higher than $\hat{i} \approx 1.5$ A, saturation of the force starts to emerge as the force no longer increases linearly in dependence of the current amplitude. An explanation for this effect resides in the influence of the magnetic field of the coil currents, which cause magnetic saturation in the soft magnetic stator. The position-dependent force variations are related to imperfections of the harmonic winding design of the two-phase coil current, as well as manufacturing and material inhomogeneities. In the derivation of analytical force factor $k_m$ in [15], simplifications of the system are applied. The difference of the analytic (18) and measured (25) force factors is attributable to the neglect of demagnetization effects of axial permanent magnets.

Summarizing, the actuator is characterized by the following differential equation of lumped parameters:

$$\begin{aligned} &m\ddot{x}_{\text{s}} - mg = F_{\text{wc}} + k_{\text{m}}\hat{i} + F_{\text{R}} \\ &m = 0.02 \text{ kg}, \ F_{\text{wc}} = 0.2 \text{ N}, \ k_{\text{m}} = 0.16 \, \tfrac{\text{N}}{\text{A}}, \\ &F_{\text{R}} = 0.21 \, \text{sgn}(\dot{x}_{\text{s}}) \text{ N}. \end{aligned} \tag{26}$$

The objective of this article is not to examine the design of a controller, but to verify the effectiveness of the weight compensation for the desired excitation profile. In [15], we presented the implementation of an extended state controller to control a similar actuator, applied here, too. A sinusoidal excitation with frequency $\omega = 5$ Hz and amplitude $\hat{x}_a = 1000$ µm of the controlled actuator is illustrated in Figure 11a. The high-frequency variation of the control signal clearly displays the non-linear influence of friction on the system.

The determining factor illustrating the effectiveness of the weight compensation is represented by the DC current component of the control signal, which is approximately zero (Figure 11a). The amplitude of the Root Mean Square (RMS) value of the current is $\hat{i}_{\text{RMS}} = 1$ A. To illustrate the effectiveness of the integrated weight compensation into the two-phase drive, the same linear drive without magnetic weight compensation is realized and comparatively evaluated. Figure 11b illustrates the position control of the linear drive without integrated weight compensation. The DC current component of the current, which is needed to compensate the static weight of the electrode, is $\hat{i}_{\text{DC}} = 1.4$ A. The maximum

current amplitude of the linear drive is restricted to $\hat{i}_{\text{max}} = 2.5$ A due to thermal limitation. This leaves a small current and force margin, which is necessary to maintain robust and high control performance for the actuator in the absence of passive weight compensation due to the allocation of resources towards static gravity compensation. Following, the control error of the actuator without passive weight compensations is significantly larger than the one with integrated compensation. Moreover, the RMS value ($\hat{i}_{\text{RMS}} = 1.6$ A) is significantly larger, illustrating the effectiveness of weight compensation.

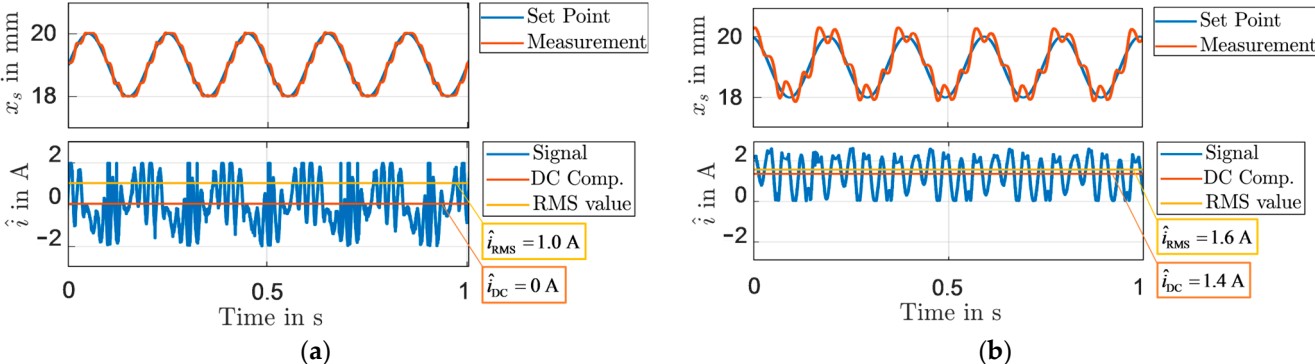

**Figure 11.** (**a**) Measurement results of the position and the current amplitude for a sinusoidal reference of the weight-compensated actuator; (**b**) measurement results of the position and the current amplitude for a sinusoidal reference of the actuator without weight compensation.

### 3.4. Discussion—Weight-Compensated Actuator for Automated EDM Process

By integrating passive permanent magnetic weight compensation into an active two-phase linear drive, it was possible to passively counterbalance the static gravity force and thereby reduce the actuator dimension. Without the integration of weight compensation, an additional dynamic force amplitude of $F_{\text{dyn}} = 0.2$ N would have been required to realize a comparatively high control quality. The actuator would have needed a significantly higher force factor $k_{\text{m}}$, which could have only been realized by increasing the number of axial permanent magnet $p$ within the slider of the active actuator, as elaborated in more detail in [15]. As illustrated in Figure 4, increasing the number of magnets in the slider increases the size and manufacturing effort of the two-phase linear drive decisively.

Finally, we briefly consider the influence of the positioning mass regarding the use of a weight-compensated actuator in the automated EDM process. The electrode rods under consideration are made of copper which possesses a density of 8.96 $\frac{\text{g}}{\text{cm}^2}$. However, graphite is also used as an electrode material for spark erosion sinking. The density of graphite is about one-fourth of that of copper, which equally reduces the positioning mass. Compensating factor $\widetilde{V}_{\text{g}}$ for the reduced mass of a graphite electrode shrinks to

$$\widetilde{V}_{\text{g}} = \frac{m_{\text{c}}g}{m_{\text{c}}\omega^2\hat{x}_a + F_R} = \frac{0.05 \text{ N}}{0.005 \text{ N} + 0.21 \text{ N}} \approx 0.23. \qquad (27)$$

The reduced positioning mass increases the relative influence of the friction and thereby significantly decreases compensating factor $\widetilde{V}$. A compensating factor of <25% does not justify the extra effort of manufacturing integrative passive weight compensation. Therefore, for the spark erosion process with graphite electrodes, the module consisting of 25 actuators does not need integrated weight compensation. The realized bundled electrode module for the EDM process with graphite electrodes is illustrated in Figure 12. This example conclusively illustrates the influence of the positioning mass on the effectiveness of magnetic weight compensation.

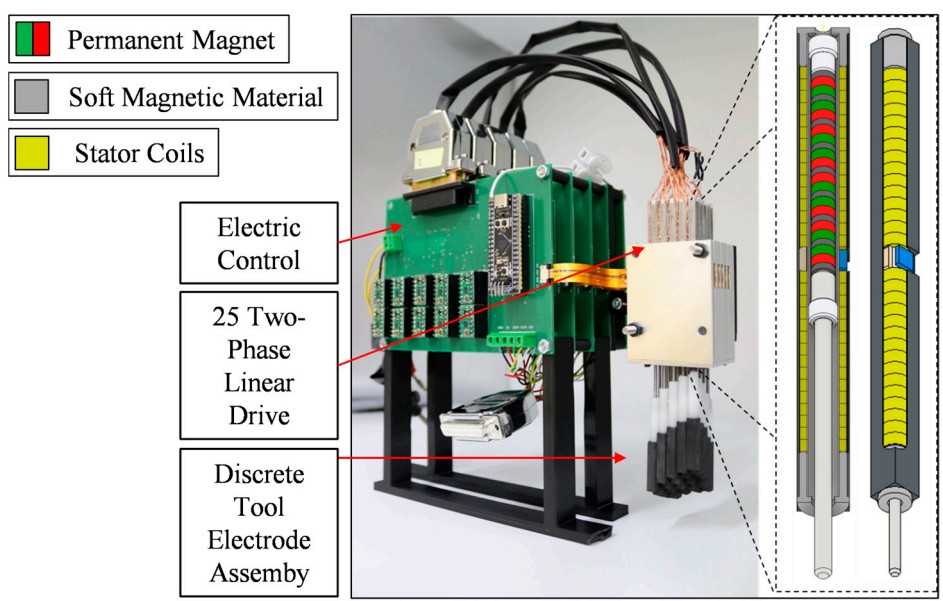

**Figure 12.** Bundled electrode module with 25 actuators for automated positioning of the graphite electrodes in the spark erosion process.

## 4. Summary

In this paper, we introduced a novel actuator concept of a two-phase linear direct drive with an integrated permanent magnetic weight compensation. The objective of passive compensation was to relieve the active actuator part and thereby enable reduced dimensioning. In this context, compensation factor *V* was introduced, which defines the ratio of the static to dynamic load and thus offers a measure of the effectiveness of passive compensation. An analytical formula for the design of force calculation of a rotationally symmetric, permanent magnetic weight compensation was derived and numerically validated. Furthermore, the analytical modeling of a two-phase linear drive and its integration with the passive compensation was illustrated. In Section 3, a weight-compensated actuator for the automated EDM process was designed, following the derived design schemes from Section 2. In this context, the influence of system parameters as positioning mass, friction, and excitation dynamics on the effectiveness of the novel actuator concept were investigated and discussed.

**Author Contributions:** B.S.W.: investigation, writing—original draft. J.M.: writing—review, supervision, funding acquisition. All authors have read and agreed to the published version of the manuscript.

**Funding:** This work was accomplished within the joined project VariSenk4EMD (project number 426311818), funded by the German Research Foundation (DFG).

**Data Availability Statement:** The data presented in this study are available on request from B. Schulte Westhoff.

**Conflicts of Interest:** The authors declare no conflicts of interest.

## Appendix A

To calculate magnetic energy, we first only consider the magnetic field of the diametrically magnetized permanent magnet with constant magnetization **M** . Neglecting axial flux is equivalent to assuming the magnet to be infinitely long, which reduces the differential equation to a two-dimensional system. The static Maxwell equations for the magnetic field **B** of the permanent magnet are given by

$$\frac{1}{\mu_0} \nabla \times \mathbf{B} = \nabla \times \mathbf{M}, \tag{A1}$$

$$\nabla \cdot \mathbf{B} = 0. \tag{A2}$$

Using the dependence of magnetic field strength **H** to magnetic field **B**,

$$\mathbf{B} = \mu_0(\mathbf{H} + \mathbf{M}), \tag{A3}$$

we reformulate (A1) and (A3) and solve for **H**:

$$\nabla \times \mathbf{H} = 0, \tag{A4}$$

$$\nabla \cdot \mathbf{H} = \nabla \cdot \mathbf{M}. \tag{A5}$$

Comparing (A5) with Gauss' law for electric charge, $\nabla \cdot \mathbf{M}$ may be interpreted as fictitious magnetic charge density $\rho_m$ representing the source for magnetic field strength **H**:

$$\nabla \cdot \mathbf{H} = \nabla \cdot \mathbf{M} = \rho_m. \tag{A6}$$

We use this analogy as a tool to solve the partial differential equation. Since the rotation of **H** is zero (A4), magnetic field strength **H** may be expressed by a potential, which we call $\phi_{\text{m}}$:

$$\mathbf{H} = \nabla \cdot \phi_{\text{m}}. \tag{A7}$$

Inserting (A6) and (A7) in (A5), we obtain the Laplace equation:

$$\Delta \phi_{\text{m}} = \rho_m. \tag{A8}$$

As shown in [20], the Laplace equation for magnetized matter can be solved using

$$\mathbf{H}(\mathbf{r}) = -(\mathbf{M} \cdot \nabla) \varepsilon(\mathbf{r}), \tag{A9}$$

where $\varepsilon(\mathbf{r})$ represents the magnetic field strength produced by the same object with a constant but fictitious magnetic charge density $\rho_M = 1$. As illustrated in [20], we use the integral form of Gauss' law to solve for $\varepsilon(\mathbf{r})$:

$$\varepsilon(\mathbf{r}) = \begin{cases} \frac{\mathbf{r}}{2} & \mathbf{r} < R_{\text{pm}} \\ \frac{\mathbf{r} A_{\text{pm}}}{2\pi r^2} & \mathbf{r} < R_{\text{pm}}. \end{cases} \tag{A10}$$

$R_{\text{pm}}$ is the radius and $A_{\text{pm}}$ the cross-section area of the permanent magnet as shown in Figure A1a. Inserting (A10) into (A9), the magnetic field strength of a cylindrical, diametrical magnetized permanent magnet calculates to

$$\mathbf{H}_{\text{pm}}(\mathbf{r}) = \begin{cases} -\frac{\mathbf{M}_{\text{pm}}}{2} & \mathbf{r} < R_{\text{pm}} \\ \frac{A_{\text{pm}}}{2\pi} \left\{ \frac{2(\mathbf{r} \cdot \mathbf{M}_{\text{pm}}) \mathbf{r}}{r^4} - \frac{\mathbf{M}_{\text{pm}}}{r^2} \right\} & \mathbf{r} > R_{\text{pm}}. \end{cases} \tag{A11}$$

The analytic solution of the magnetic field strength for an exemplary magnetization is illustrated in Figure A1a. Next, we consider the analytic solution for a cross-section including the soft magnetic stator (Figure A1c). Given the assumption of an ideal magnetic conductor, we seek for a solution of the partial differential equation in which the magnetic field strength within the soft magnetic stator vanishes. The boundary condition at the inner soft magnetic surface is given by

$$\mathbf{n} \times \mathbf{H}(\mathbf{r}_{\text{sm}}) = 0 \text{ with } |\mathbf{r}_{\text{sm}}| = R_{\text{sm}}. \tag{A12}$$

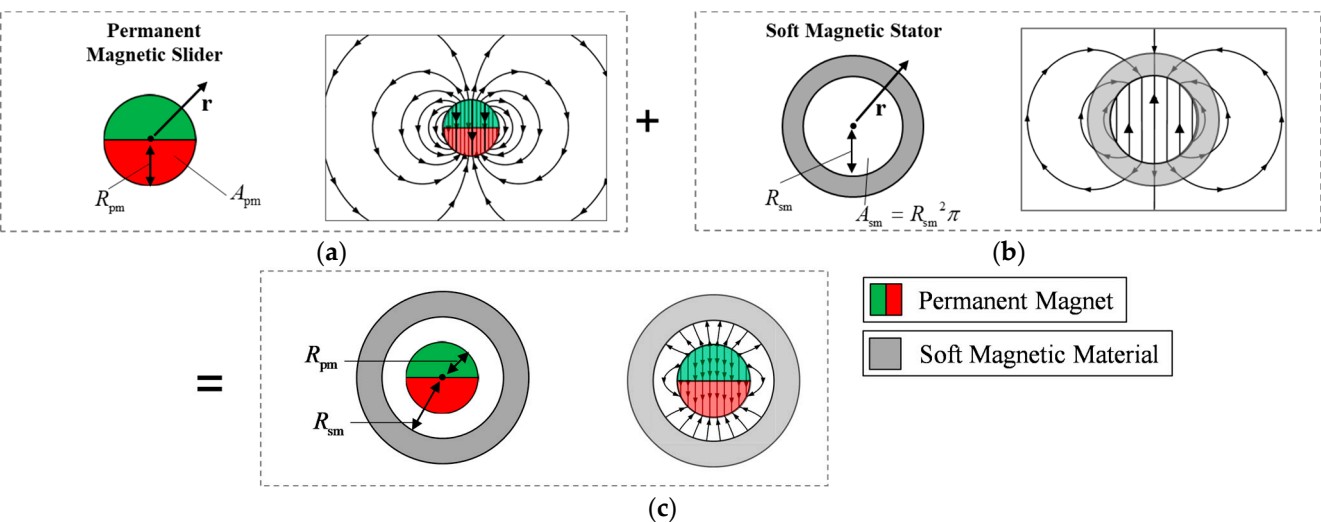

**Figure A1.** (**a**) Magnetic field strength of a diametrically magnetized permanent magnet. (**b**) Resultant magnetic field strength of the soft magnetic stator. (**c**) Magnetic field of the actuator cross-section of magnetic weight compensation.

In other words, the superposition of the permanent magnetic field strength (Figure A1a) and the soft magnetic field strength (Figure A1b) cancels out in the soft magnetic stator. We propose the distribution of the magnetic field of soft iron $\mathbf{H}_{sm}$ as follows:

$$
\mathbf{H}_{sm}(\mathbf{r}) =
\begin{cases}
-\dfrac{\mathbf{M}_{sm}}{2} & \mathbf{r} < R_{sm} \\
\dfrac{A_{sm}}{2\pi} \left\{ \dfrac{2(\mathbf{r} \cdot \mathbf{M}_{sm})\mathbf{r}}{r^4} - \dfrac{\mathbf{M}_{sm}}{r^2} \right\} & \mathbf{r} > R_{sm}.
\end{cases}
\tag{A13}
$$

Equation (A13) has the same form as (A11), but different magnetization $\mathbf{M}_{sm}$ and cross-section area $A_{sm}$. By choosing magnetization $\mathbf{M}_{sm}$ to be

$$
\mathbf{M}_{sm} = -\mathbf{M}_{pm} \frac{A_{pm}}{A_{sm}} = -\mathbf{M}_{pm} \frac{R_{pm}{}^2}{R_{sm}{}^2},
\tag{A14}
$$

the superposition of the fields in the soft magnetic stator just cancel out, satisfying boundary condition (A12). Inserting (A14) in (A13) and summing with (A11) yields the distribution of the fields as illustrated in Figure A1c:

$$
\mathbf{H}(\mathbf{r}) =
\begin{cases}
-\dfrac{\mathbf{M}_{pm}}{2} + \dfrac{\mathbf{M}_{pm}}{2} \dfrac{A_{pm}}{A_{sm}} & \mathbf{r} < R_{pm} \\
\dfrac{A_{pm}}{2\pi} \left\{ \dfrac{2(\mathbf{r} \cdot \mathbf{M}_{pm})\mathbf{r}}{r^4} - \dfrac{\mathbf{M}_{pm}}{r^2} \right\} & R_{sm} > \mathbf{r} > R_{pm} \\
\quad + \dfrac{\mathbf{M}_{pm}}{2} \dfrac{A_{pm}}{A_{sm}} & \\
0 & \mathbf{r} > R_{sm}.
\end{cases}
\tag{A15}
$$

To derive the force by the variation of energy, the magnetic energy of weight compensation needs to be calculated. The brought-in energy to magnetize matter is given by [21]

$$
U_m = \int\limits_V \int\limits_0^{\mathbf{B}} \frac{1}{\mu_0} \mathbf{H} d\mathbf{B} \, dV.
\tag{A16}
$$

For permanent magnetic material, the relationship between $\mathbf{H}$ and $\mathbf{B}$ is nonlinear and hysteretic. The total energy brought in to magnetize a permanent magnet is thus not the same as the saved field energy. Energy dissipates into heat when magnetization $\mathbf{M}$ changes. However, for the application of weight compensation, we consider magnetization $\mathbf{M}$ to be constant such that no energy dissipates in operation. Therefore, we may use (A16) to

compute the force by variation of energy. Considering **M** to be constant, the relationship between **H** and **B** is affine linear:

$$\mathbf{H} = \frac{1}{\mu_0}\mathbf{B} - \mathbf{M}. \tag{A17}$$

Since we neglect axial flux, Equation (A16) may be reformulated to

$$U_\mathrm{m} = \sum_i l_i \widetilde{U}_i, \tag{A18}$$

where $l_i$ denotes the axial length and $\widetilde{U}_i$ is the energy per length of a cross-section area $A_i$ (surface energy):

$$\widetilde{U}_i = \int\limits_{A_i} \int\limits_0^{\mathbf{B}} \frac{1}{\mu_0}\mathbf{H}d\mathbf{B}\, dA. \tag{A19}$$

Following Figure 1, the energy of weight compensation formulates to

$$U_\mathrm{wc} = \widetilde{U}_\mathrm{A}(l_0 - x_s) + \widetilde{U}_\mathrm{B}x_s. \tag{A20}$$

In order to calculate $\widetilde{U}_i$, we first subdivide the integral into the surface energy of permanent magnet field $\widetilde{U}_{i,\mathrm{pm}}$, of soft magnetic stator field $\widetilde{U}_{i,\mathrm{sm}}$ and mutual surface energy $\widetilde{U}_{i,\mathrm{mutual}}$ of the superimposed fields:

$$\widetilde{U}_i = \widetilde{U}_{i,\mathrm{pm}} + \widetilde{U}_{i,\mathrm{sm}} + \widetilde{U}_{i,\mathrm{mutual}}. \tag{A21}$$

To calculate $\widetilde{U}_{i,\mathrm{pm}}$, Equation (A17) is inserted in (A19):

$$\begin{aligned} \widetilde{U}_{i,\mathrm{pm}} &= \int\limits_{A_i} \int\limits_0^{\mathbf{B}_\mathrm{pm}} \left(\frac{1}{\mu_0}\mathbf{B} - \mathbf{M}_\mathrm{pm}\right)d\mathbf{B}\, dA \\ &= \int\limits_{A_i} \left(\frac{1}{2\mu_0}\mathbf{B}_\mathrm{pm}\cdot\mathbf{B}_\mathrm{pm} - \mathbf{M}_\mathrm{pm}\cdot\mathbf{B}_\mathrm{pm}\right)\, dA. \end{aligned} \tag{A22}$$

In the later following force calculation, we differentiate (A22) with respect to slider position $x_\mathrm{s}$. Since magnetization $\mathbf{M}_\mathrm{pm}$ is constant and thereby independent of $x_\mathrm{s}$, we add

$$c = \int \frac{\mu_0 \mathbf{M}_\mathrm{pm}\cdot\mathbf{M}_\mathrm{pm}}{2}\, dV \tag{A23}$$

to (A22) without changing the resultant force. Adding (A23) to (A22) and using (A17) results in

$$\widetilde{U}_{i,\mathrm{pm}} = \int\limits_{A_i} \left(\frac{\mu_0}{2}\mathbf{H}_\mathrm{pm}\cdot\mathbf{H}_\mathrm{pm}\right)\, dA. \tag{A24}$$

To calculate (A24) considering the derived field distribution (A11), we first simplify the integral. The energy of free stationary currents $U_\mathrm{F}$ is given by [22]

$$U_\mathrm{F} = \int\limits_V \frac{1}{2}\mathbf{H}\cdot\mathbf{B}\, dV. \tag{A25}$$

The regarded system has no free currents, which implies that **H** and **B** are orthogonal vector fields:

$$0 = \int\limits_V \frac{1}{2}\mathbf{H}\cdot\mathbf{B}\, dV. \tag{A26}$$

Inserting (A17) in (A26) shows that

$$\int_V \frac{\mu_0}{2} \mathbf{H} \cdot \mathbf{H} \, dV = -\int_V \frac{\mu_0}{2} \mathbf{H} \cdot \mathbf{M} \, dV. \tag{A27}$$

Using (A27) and (A11) to calculate (A24) results in

$$
\begin{aligned}
\widetilde{U}_{i,\mathrm{pm}} &= \int_{A_i} \left( -\frac{\mu_0}{2} \mathbf{H}_{\mathrm{pm}} \cdot \mathbf{M}_{\mathrm{pm}} \right) dA \\
&= \int_{A_i} \mu_0 \frac{M_{\mathrm{pm}}^2}{4} \, dA = \mu_0 \frac{M_{\mathrm{pm}}^2}{4} A_{\mathrm{pm}}.
\end{aligned}
\tag{A28}
$$

For the soft magnetic stator, we use (A13) to calculate surface energy. The same proceeding leads to

$$
\begin{aligned}
\widetilde{U}_{\mathrm{sm},i} &= \int_{A_i} \left( -\frac{\mu_0}{2} \mathbf{H}_{\mathrm{sm}} \cdot \mathbf{M}_{\mathrm{sm}} \right) dA \\
&= \mu_0 \frac{M_{\mathrm{sm},i}^2}{4} A_{\mathrm{sm},i} = \mu_0 \frac{M_{\mathrm{pm}}^2}{4} \frac{A_{\mathrm{pm}}^2}{A_{\mathrm{sm},i}}.
\end{aligned}
\tag{A29}
$$

The mutual surface energy results from the overlapping field terms:

$$\widetilde{U}_{i,\mathrm{mutual}} = -\int_{A_i} \frac{\mu_0}{2} \mathbf{H}_{\mathrm{pm}} \cdot \mathbf{M}_{\mathrm{sm},i} \, dA - \int_{A_i} \frac{\mu_0}{2} \mathbf{H}_{\mathrm{sm}} \cdot \mathbf{M}_{\mathrm{pm},i} \, dA. \tag{A30}$$

The two terms of the mutual energies are of the same value [23] ($\nabla \times \mathbf{M}$ may be interpreted as a magnetization current). Doubling the second term of (A30) and inserting (A13) and (A14) yields

$$
\begin{aligned}
\widetilde{U}_{i,\mathrm{mutual}} &= -\int_{A_i} \mu_0 \mathbf{H}_{\mathrm{sm},i} \cdot \mathbf{M}_{\mathrm{pm}} \, dA = \int_{A_i} \frac{\mu_0}{2} \mathbf{M}_{\mathrm{sm},i} \cdot \mathbf{M}_{\mathrm{pm}} \, dA \\
&= \frac{\mu_0}{2} \mathbf{M}_{\mathrm{pm}} \cdot \mathbf{M}_{\mathrm{sm},i} A_{pm,i} = -\frac{\mu_0}{2} M_{\mathrm{pm}}^2 \frac{A_{pm}^2}{A_{\mathrm{sm},i}} \, .
\end{aligned}
\tag{A31}
$$

Summing (A28), (A29) and (A31) finally results in surface energy:

$$
\begin{aligned}
\widetilde{U}_i &= \widetilde{U}_{i,\mathrm{pm}} + \widetilde{U}_{i,\mathrm{sm}} + \widetilde{U}_{i,\mathrm{mutual}} \\
&= \mu_0 \frac{M_{\mathrm{pm}}^2}{4} A_{\mathrm{pm}} \left( 1 - \frac{A_{\mathrm{pm}}}{A_{\mathrm{sm},i}} \right).
\end{aligned}
\tag{A32}
$$

The total energy is calculated by multiplying the corresponding lengths of the two different cross-sections, $A_{\mathrm{A}}$ and $A_{\mathrm{B}}$ (Figure 1):

$$U_{\mathrm{wc}} = \mu_0 \frac{M_{\mathrm{pm}}^2}{4} A_{\mathrm{pm}} \left( \left( 1 - \frac{A_{\mathrm{pm}}}{A_{\mathrm{sm, A}}} \right) x_s - \left( 1 - \frac{A_{\mathrm{pm}}}{A_{\mathrm{sm, B}}} \right) (l_0 - x_s) \right). \tag{A33}$$

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
