# Peer review of "Design of an Electromagnetic Linear Drive with Permanent Magnetic Weight Compensation"

_actuators, doi:10.3390/act13030107_

Round 1

Reviewer 1 Report

Comments and Suggestions for Authors

(1) The design and implementation of the magnetic compensation mechanism is not discussed in detail. The integration of passive permanent magnet gravity compensation into an active two-phase linear actuator is mentioned in the paper, but no detailed design and implementation details are provided.

(2) Lack of performance evaluation of the magnetic compensation mechanism. The paper mentions that the size of the actuator is reduced and the control quality is improved by magnetic compensation, but no specific performance evaluation data or comparative results are provided.

(3) There is a lack of discussion of system stability and reliability. The paper does not discuss the impact of the magnetic compensation mechanism on the stability and reliability of the system, such as the life of the magnetic compensation mechanism, the failure rate, and the immunity to external disturbances.

(4) Lack of comparison with other possible gravity compensation mechanisms, the paper only discusses the method to reduce the size of the drive by permanent magnet gravity compensation, but does not compare with other possible gravity compensation mechanisms. Suggestion and addition of information on comparison with other compensation mechanisms.

Comments on the Quality of English Language

Overall the paper is well-written.

Reviewer 2 Report

Comments and Suggestions for Authors

Dear authors, once in a while I get manuscripts to review which are a pleasure to read. Your manuscript is one of those. Thank you! You paid attention to details, the structure of the manuscript is very logical, and your formulations are clear. I appreciate your honest discussion of the significance of your design. Your manuscript is an example for colleagues! You may want to correct the "magntic"  in figure 4, and "ration" on line 240, but that's it from my side.

Author Response

Thank you very much for taking the time to review this manuscript and this very nice respond! We corrected the mentioned typos.

Reviewer 3 Report

Comments and Suggestions for Authors

The paper proposes using permanent magnet compensation of gravity so that  the electromagnetic control can be smaller.  The authors then develop the fairly obvious fact that permanent magnets can generate a constant force and if the permanent magnet flux is far enough from the electromagnetic flux then there is little interference. But the authors do not sufficiently justify their approach nor compare their approach to other methods. Magnetic bearings use a constant current to generate a lifting force and then control that current above and below the steady state to accomplish the dynamic requirement. This type of control should be demonstrated and compared to the authors' proposal.  A passive spring-like element is the cheapest way to compensate gravity but, as pointed out by the authors, a spring does have stroke limitations. But certainly one can design the system so the spring moves with the control load and always counters gravity. This is not intended to be design proposals. It is pointing out that more justification is needed for the use of permanent magnet gravity compensation over other methods that would appear to be better or cheaper or work just as well and so forth.

Comments on the Quality of English Language

The English is fine. There is always things that could be stated better.

Reviewer 4 Report

Comments and Suggestions for Authors

Could the following points be reconsidered?

1) There have been studies on using voice coil motors and reluctance electromagnetic actuators to drive an electrode in electric discharge machines. Furthermore, some studies use magnetic levitation/bearing mechanisms to support the electrode and reduce friction, which is the problem in this paper. This should be investigated and included in the references.

2)The novelty of this research lies in the fact that the proposed actuators are integrated into a matrix. Shouldn't magnetic interference be considered in the analysis due to leakage flux from each actuator?
